# Bioactive Compounds and Antioxidant Properties of Wild Rocket (*Diplotaxis Tenuifolia* L.) Grown under Different Plastic Films and with Different UV-B Radiation Postharvest Treatments

**DOI:** 10.3390/foods11244093

**Published:** 2022-12-17

**Authors:** Raffaele Romano, Fabiana Pizzolongo, Lucia De Luca, Eugenio Cozzolino, Massimo Rippa, Lucia Ottaiano, Pasquale Mormile, Mauro Mori, Ida Di Mola

**Affiliations:** 1Department of Agricultural Sciences, University of Naples Federico II, 80055 Portici, Italy; 2Council for Agricultural Research and Economics (CREA)—Research Center for Cereal and Industrial Crops, 81100 Caserta, Italy; 3National Research Council (CNR)—Institute of Applied Sciences and Intelligent Systems “Eduardo Caianiello”, 80128 Napoli, Italy

**Keywords:** rocket, UV-B radiation, plastic films, bioactive compounds, phenolic compounds

## Abstract

Rocket species are rich in nutrients with well-known bioactive activity, but their content depends on several factors, such as plant–UV radiation interaction. In this work, we measured the production of nutritional elements in wild rocket (*Diplotaxis tenuifolia* L.) leaves as a function of exposure to UV-B radiation by adopting a combined approach. The wild rocket plants were grown under three greenhouse cover films (A, B, and C) having different transmittivity to UV-B and the fresh-cut leaves were exposed to UV-B in postharvest for 45, 150, 330, and 660 s. The content of chlorophyll, carotenoids, phenolic compounds, ascorbic acid, and the antioxidant activity were determined. Chlorophyll, carotenoids, and total phenolic content were significantly increased by the combination of Film C and treatment with UV-B for 45 s. The predominant phenolic compounds were kaempferol, isorhamnetin, and quercetin. Film C also elicited an increase in ascorbic acid (the most abundant antioxidant compound in the range 374–1199 per 100 g of dry matter) and antioxidant activity. These findings highlighted an increase in bioactive compound content in the wild rocket when it was cultivated under Film C (diffused light film with a tailored UV-B transmission dose) and treated with UV-B radiation for 45 s postharvest, corresponding to an energy dose of 0.2 KJ m^−2^.

## 1. Introduction

Wild rocket (*Diplotaxis tenuifolia* L.) is a green leafy vegetable belonging to the Brassicaceae family. It has a pleasantly bitter taste and contains beneficial phytonutrients with antioxidant activity (vitamin C, carotenoids, phenols) useful for human health [1,2,3,4,5]. Rocket leaves are commonly consumed in the Mediterranean diet as fresh vegetables or, after washing and packaging, as ready-to-eat products.

Previous studies have shown that rocket species are rich in polyphenolic compounds with biological effects that have been extensively studied and reviewed in the literature [6,7,8] and also have an important role in determining the characteristic flavour [9]. Phenolic compounds (phenolic acids, simple phenols, hydroxycinnamic acid derivatives, and flavonoids) are potentially protective factors against heart diseases and cancer, in part because of their potent antioxidative properties [9,10].

Quercetin and kaempferol derivatives were found in rocket by Pasini et al. [9] with quercetin as the main phenolic compound of *Diplotaxis tenuifolia* and kaempferol of the *Eruca sativa* cultivar. Rutin and isorhamnetin were also found [11]. Durazzo et al. [12] pointed out the characteristics of wild rocket as a dietary source of antioxidants, reporting a significant effect of conventional and integrated cultivation practices on product quality and no effect on biological activity.

The wild rocket is also rich in ascorbic acid (vitamin C), a metabolite with antioxidant activity and an important cofactor for enzymes. Humans do not synthesize vitamin C, so it is important to integrate this compound into food [13].

Many studies have demonstrated that the phenolic content depends on ultraviolet-B (UV-B) radiation treatments and the covering films of the greenhouse [14,15].

UV-B radiation treatments are an emerging technology that may be used to generate fruits, vegetables, and herbs enriched with secondary plant metabolites that are very beneficial for human health. UV-B radiation has been shown to affect growth, photosynthesis, secondary plant metabolites, and plant–insect interactions in various horticultural and agricultural crops [16]. Plants have evolved different protective mechanisms against environmental stresses, such as UV-B radiation, which generates highly reactive oxygen species. Agarwal et al. [17] found that *Cassia auriculata* L. seeds try to contrast high concentrations of the oxygen species, cytotoxic and highly reactive, produced under UV-B radiation by increasing the antioxidant activity and the content of antioxidant compounds. Caldwell et al. [18] also found that UV-B radiation increased the carotenoid and chlorophyll concentrations of greenhouse-grown green leaf lettuce. Higashio et al. [19] reported that UV radiation after harvest improves the quality of strawberries and onions and, in particular, the content of quercetin in onion doubled. Kasim and Kasim [20] showed that the treatment of fresh-cut spinach (*Spinacia oleracea* L.) with UV-B reduced leaf yellowing and improved the visual quality of the sample during storage. Recently, Hao et al. [21] showed that UV-B supplementation positively influenced the accumulation of bioactive compounds in pak choi (*Brassica rapa* L.).

UV-B treatment influences the carotenoid content in plants, but this effect depends on the specific carotenoid compound present in plants [14]. In this way, canola seedlings (*Brassica napus*) grown under ambient (5 kJ m^−2^ d^−1^) and enhanced UV-B radiation conditions were shown to produce generally higher concentrations of total carotenoids than those grown under zero UV-B radiation [22]. Furthermore, in tomato of the Liberto cultivar, a preharvest application of UV-B increased the lycopene and ß-carotene content [23].

Recently, increasing attention has been given to the effect of ultraviolet light on the preservation of postharvest fruits and vegetables. Existing studies have shown that ultraviolet light can improve the quality of various postharvest fruits and vegetables [24,25]. UV-B is UV light that plants can be exposed to under natural conditions but with less damage than UV-C; thus, it is a more promising method for postharvest storage of fruits and vegetables [26]. A recent study reported applications of UV-A, -B, and -C radiation in the postharvest storage of fruits and vegetables, which include the effect on disease occurrence, phenolic metabolism, and important quality indicators, such as ethylene production, respiration rate, firmness, chlorophyll metabolism, enzymatic antioxidant system, and nonenzymatic antioxidant system [27]. In this review article, the authors also present an outlook for the further application of UV on the postharvest storage of fruits and vegetables.

Postharvest UV-B exposure was also found to increase carotenoid biosynthesis in tomato fruits by a low UV-B dosage of 0.54 kJ m^−2^ d^−1^ [28,29]. The accumulation of carotenoids in tomato has been observed to be linked to both ethylene-dependent and ethylene-independent mechanisms [30]. Moreover, leaf carotenoid content in tobacco (*Nicotiana tabacum* L., ‘K326’) has also been shown to be able to be increased with low UV-B exposure (9.75 μW/cm^2^) [31].

Despite the very extensive literature on this topic, in all these works, only treatments based on natural or artificial radiation are taken into consideration to study the modifications of the most significant substances of the plant species examined induced by UV-B, highlighting a lack of studies in which a combined approach is conducted.

In this work, we measured the content of chlorophylls, carotenoids, phenolic compounds, and vitamin C and the antioxidant activity of wild rocket leaves first grown under three greenhouse cover films with both different optical properties and UV-B transmittance and subsequently treated postharvest with artificial UV-B radiation with different exposure times. As far as we know, it is the first time that this combined approach, based on the use of both natural and artificial UV-B radiation, has been tested to increase the nutraceutical properties of foods. This further deepening has been conducted to complete, not only the study on the behaviour of the rocket salad irradiated with an artificial UV-B light source in the fresh-cut phase, but also to select a way to treat the on-line salad, further enriched in nutrients, for the market.

## 2. Materials and Methods

### 2.1. Experimental Design and Growing Conditions

The experiment was carried out during autumn 2020 and spring 2021 at Gussone Park in Portici (Naples, Italy), the experimental site of the Department of Agricultural Science.

The cultivar ‘Reset’ (Maraldi Sementi Srl, Cesena, Italy) of wild rocket (*Diplotaxis tenuifolia* L.) was transplanted under three greenhouses covered with plastic films characterized by different optical properties: 1. Film A (150-microns thick), a diffused light film; Film B (150-microns thick), a clear plastic film; Film C (150-microns thick), a diffused light film with a tailored UV-B transmission dose.

The rocket variety had green leaves with medium-sized lobes and high yield. In addition, the rocket variety had good tolerance to Fusarium spp. and great growing flexibility, making it suitable for production in any season [32]. The agricultural practices were ordinary for the cultivation area. Only nitrogen was applied at a rate of 18 kg ha^−1^ per cut as ammonium nitrate (34%). Water losses were calculated by the Hargreaves formula and completely restored. No pesticide treatments were used.

The experimental design was a split-plot that compared the greenhouse cover film as the main factor and five postharvest treatments with UV-B as the secondary factor; all treatments were replicated three times.

The five UV-B treatments were as follows: 1. not treated; 2. treated for 45 s; 3. treated for 150 s; 4. treated for 330 s; and 5. treated for 660 s, hereafter, referred to as I, II, III, IV, and V, respectively.

A fluorescent tube (Philips TL20 W/12), positioned 20 cm from rocket leaves, was used for UV-B irradiation. UV-B intensity was measured and controlled using the photoradiometer Delta Ohm HD 2102.2 with a probe in the spectral range 280–315 nm. Each treatment corresponded to five energy doses: 0, 0.2, 0.7, 1.5, and 3.0 KJ m^−2^ for I, II, III, IV, and V, respectively.

Under Film A and Film B, the transplant was made on 8 October in 0.38-m^2^ pots filled with sandy soil (91% sand, 4.5% silt, and 4.5% clay, USDA classification), with 253 ppm P_2_O_5_, 490 ppm K_2_O, 2.5% organic matter, 0.101% total nitrogen, and pH 7.4.

Under Film C, the transplant was similarly made on 8 October directly in the soil, which had the following characteristics: clay loam soil (35.5% sand, 25.5% silt, and 39.0% clay, USDA classification), with 93.3 ppm P_2_O_5_, 750.3 ppm K_2_O, 1.7% organic matter, 0.09% total nitrogen, and pH 7.6.

Under all three greenhouses, the wild rocket was harvested six times, and at the harvest of April 8, a 30 g sample of leaves for each treatment was used for UV-B applications. Then, the samples were oven-dried at 60 °C until reaching a constant weight to determine the dry matter percentage.

### 2.2. Plastic Film Optical Properties

Film A with 58% diffusivity is marketed with the trade name “Sunsaver Diff”, manufactured by the Israeli company Ginegar Plastic Products. The main characteristics of this film are antidrip action, 87% thermicity, and a total transmittivity (direct plus diffused components of light transmitted) in photosynthetically active radiation (PAR) of 90%. Film B is marketed with the commercial name “Lirsalux” by Lirsa Srl (Ottaviano, NA, Italy). Film B has an antidrip effect, 75% thermicity, no transmission in the UV-B range, and a total transmittivity in PAR of 85%. More details on the two films are reported in Di Mola et al. [33]. Film C is the same as Film A, but it exhibits a “window” in the UV-B range (280–315 nm) of approximately 30% of the total UV-B radiation belonging to the solar radiation.

### 2.3. Chemical Analysis

Dry matter content, chlorophyll a, b, and total, and carotenoids were determined in fresh rocket leaves. Total and individual polyphenols, ascorbic acid and antioxidant activity assays were performed on lyophilized rocket leaves. Lyophilization was performed at 50 °C and <0.05 mbar for 48 h in a freeze dryer (LIO-5PDGT, Cinquepascal s.r.l., Trezzano sul Naviglio, (MI), Italy).

All solvents and reagents used for experiments were purchased from Sigma-Aldrich Co. (Milano, Italy).

#### 2.3.1. Dry Matter Content

The determination of the dry matter, which represents the total solids, was carried out following the method of Luca et al. [34], with modifications. Briefly, 5 g of rocket leaves was dried in an oven at 70 °C until the complete elimination of water that occurred in 24 h. The results were expressed as a weight/weight percentage of dry matter (% *w*/*w*).

#### 2.3.2. Chlorophyll a, b, and Total Carotenoids

These determinations were performed with the spectrophotometric method described by Gutiérrez et al. [35], with some modifications. Approximately 400 mg of lyophilized rocket leaves was dissolved in 15 mL of a mixture of acetone/water (80:20), ground using an IKA T-25 Ultra Turrax homogenizer (Staufen, Germany) at 5000 rpm for 15 min and filtered. The solution was placed in a quartz cuvette, and absorbance measurements were performed by using a UV-1601 spectrophotometer (Shimadzu, Kyoto, Japan) at wavelengths of 663.2, 646.8, and 470 nm. The equations described by Gutiérrez et al. [35] were used to determine the individual levels of chlorophyll a (Ca = 12.25 A663.2−2.79 A646.8), chlorophyll b (Cb = 521.5 A646.8−5.1 A663.2), and total carotenoids [Cx + c = (1000 A470−1.82 Ca−85.02 Cb)/198]. The results were expressed as mg/100 g of dry weight (D.W.).

#### 2.3.3. Total Polyphenol Content

The total polyphenol content (TPC) of rocket leaves was determined by the Folin-Ciocalteu method reported by Toledo-Martín et al. [36], with modifications. Briefly, 400 mg of lyophilized rocket leaves was added to 8 mL of ethanol (with 1% hydrochloric acid) and shaken for 30 s and centrifuged at 4000× *g* for 15 min. To 180 μL of the supernatant, 300 μL of Folin-Ciocalteu reagent, 4800 μL of deionized H_2_O, and 900 μL of a 20% Na_2_CO_3_ solution in water (*w*/*v*) were added. After incubation for 60 min in the dark at room temperature, the absorbance was read at 765 nm using a UV-1601 UV-Visible spectrophotometer (Shimadzu, Milan, Italy). A calibration curve (*R*^2^ = 0.99) was constructed with gallic acid at different concentrations (25, 50, 100, 150, 200 mg/L). The results were expressed as mg of gallic acid equivalent (GAE)/100 g of dry weight (mg GAE/100 g D.W.).

#### 2.3.4. Individual Phenolic Compounds

Phenolic compounds were determined as described by Martínez-Sánchez [37]. A quantity of 500 mg of lyophilized rocket leaves was added to 20 mL of deionized water/methanol (50:50), ground using an IKA T-25 Ultra Turrax homogenizer (Staufen, Germany) at 12,000× *g* for 30 s, centrifuged at 4000× *g* for 10 min and filtered. A quantity of 20 μL was injected into a HPLC system (Agilent 1100 Series, Santa Clara, CA, USA) equipped with degaser G4225A, diode array detector G1315B, and a Spherisorb ODS2 (5 μm, 4.6 mm × 250 mm) C18 reversed-phase column following the HPLC method proposed by Romano et al. [38]. The mobile phases were composed of 0.1% formic acid in water (phase A) and acetonitrile (phase B). The elution gradient was as follows: 0–5 min, 10% B; 5–10 min, 15% B; 10–16 min, 15% B; 16–18 min, 18% B; 18–28 min, 30% B; 28–33 min, 40% B; 33–35 min, 50% B; 35 min, return to initial conditions. The flow was set at 1.0 mL min^−1^. Phenolic detection was performed at 260 nm (kaempferol, isorhamnetin, quercetin, and rutin), 280 nm (gallic acid) and 330 nm (caffeic acid). Typical chromatograms of standards and sample are reported in Appendix A. To quantify the concentration of compounds, calibration curves of standards (kaempferol, isorhamnetin, quercetin, rutin, caffeic acid, gallic acid) were constructed in the range 0–500 ppm. The limit of detection (LOD) and limit of quantification (LOQ) were 0.5 and 1 ppm, respectively, for kaempferol, caffeic, and gallic acid, 1 and 2 ppm, respectively, for isorhamnetin and quercetin, and 2 and 4 ppm for rutin. The results were expressed as mg of the phenolic compound/100 g of D.W.

#### 2.3.5. Ascorbic Acid

The determination of the ascorbic acid was performed by titration as reported by Ragusa et al. [39], with modifications. The method is based on the reduction of 2,6-dichlorophenol-indophenol by ascorbic acid. Briefly, 100 mg of lyophilized rocket leaves was added to 10 mL of 3% metaphosphoric acid, ground using an IKA T-25 Ultra Turrax homogenizer at 12,000× *g* for 60 s, centrifuged at 6500× *g* for 10 min and filtered. A quantity of 1 mL of this extract was added to 10 mL of deionized water and titrated with 1.5 mM 2,6-dichlorophenol-indophenol. The ascorbic acid content was quantified by comparison with a standard calibration curve obtained with known concentrations of ascorbic acid in pure methanol solution (0, 5, 20, 50, and 100 ppm). The results were expressed as mg of ascorbic acid/100 g of dry weight (mg AA/100 g D.W.).

#### 2.3.6. Antioxidant Activity

The antiradical activity of the rocket leaves was determined by ABTS and DPPH assays according to Romano et al. [40] and Araújo-Rodrigues et al. [41], with modifications. A quantity of 1 g of lyophilized rocket leaves was added to 20 mL 80% methanol, ground using an IKA T-25 Ultra Turrax homogenizer at 12,000× *g* for 30 s, centrifuged at 5000× *g* for 10 min and filtered.

In the ABTS assay, 1800 μL of ABTS working solution was added to 200 μL of extract dissolved in methanol. The solution was kept in the dark for 5 min, and the absorbance was measured at 734 nm using a UV-1601 UV-Visible spectrophotometer. The antiradical activity was calculated using a Trolox calibration curve with different concentrations (25, 50, 75, 100, 125, 150, 175, and 250 μM). The results were expressed as mg Trolox equivalent (TE)/100 g D.W.

In the DPPH assay, to 250 mL of extract dissolved in methanol, 1750 mL of 0.1 mM DPPH working solution was added. The solution was kept in the dark for 30 min, and the absorbance was measured at 515 nm using a UV-1601 UV-Visible spectrophotometer. The antiradical activity was calculated using a Trolox calibration curve with different concentrations (25, 50, 75, 100, 125, 150, 175, and 250 μM). The results were expressed as mg TE/100 g D.W.

#### 2.3.7. Statistical Analysis

All analytical determinations were repeated three times. The data were subjected to analysis of variance (ANOVA) with the SPSS software package (SPSS version 22, Chicago, IL, USA); the means were separated using Tukey’s test at *p* ≤ 0.05.

## 3. Results and Discussion

### 3.1. Dry Matter Percentage of Rocket Leaves

The dry matter percentage of rocket leaves was found to be in the range of 8.29–9.94% *w*/*w* (Figure 1), findings similar to the findings of Caruso et al. [42], while Schiattone et al. [43] showed a value that ranged between 9.0 and 10.9% *w*/*w*, depending on salinity positively influencing the dry matter content. In general, leaves with lower moisture (and therefore with higher dry matter) maintain better organoleptic characteristics during storage. The dry matter percentage was only statistically affected by greenhouse cover films; neither UV-B applications nor the interaction of the two factors was significant (Figure 1). Notably, Film A and Film C showed the highest value (8.95% *w*/*w* on average) with a 4.1% increase compared to the clear film (Film B); however, only Film C was significantly different from Film B.

### 3.2. Chlorophyll a, Chlorophyll b, and Carotenoid Content of Rocket Leaves

The interaction between greenhouse cover films and UV-B application affected the pigment content (Figure 2, Figure 3 and Figure 4). Chlorophyll a, b, and carotenoid content were found in the range of 485.75–997.24, 211.70–652.15, and 69.10–256.81 mg/100 g D.W., respectively (Figure 2, Figure 3 and Figure 4) and were highest in rocket transplanted under Film C. Similar values were found by Gutiérrez, et al. [35] in minimally processed rocket. Martínez-Sánchez et al. [37] showed a concentration that ranged from 30 to 120 mg/100 g of fresh weight for chlorophyll a, while the chlorophyll b content ranged from 15 to 40 mg/100 g of fresh weight depending on sanitizer agents used for controlling microbial growth, time, and type of storage. The total chlorophyll content is associated with the retention of green colour, one of the most general indices used to evaluate the overall quality and freshness of green leafy vegetables. Moreover, carotenoids and chlorophylls have an important role in the prevention of various diseases associated with oxidative stress, such as cancer, cardiovascular diseases, and other chronic diseases [44]. Film C always elicited higher values of all three parameters; in particular, chlorophyll a elicited a 17.0% and 48.6% increase over Film B and Film A, respectively (Figure 2). The values ranged between 506 of Film A-I (no application of UV-B) and 975.9 mg per 100 g of D.W. of Film C-II (applications of UV-B for 45 s). Additionally, for Film A and Film B, the treatment with UV-B for 45 s elicited higher values of chlorophyll and was different from the other treatments of each film.

The effect of Film C on chlorophyll b was stronger, with values of 49.0% and 102.0% more than Film B and Film A, respectively. For this parameter, both UV-B treatments for 45 s and 150 s, corresponding to energy doses of 0.2 and 0.7 KJ m^−2^, respectively, elicited the highest values, which were significantly different from all other treatments (Figure 3). Additionally, under Film A, the same treatments were not different and elicited higher values. Instead, under Film B, only the UV-B treatment for 150 s was allowed to reach higher values and was not different from treatments I and V of Film C. Again, the lowest value was recorded in rocket grown under Film A and not treated with UV-B.

Finally, for the carotenoid content, the effect of Film C was more marked, with a 25.7% and 78.9% increase compared to Film B and Film A, respectively (Figure 4). Once again, the highest value was recorded in the rocket grown under Film C and treated with UV-B for 45 s, and the lowest was recorded in rocket plants grown under Film A and not treated with UV-B. For this parameter, under both Film A and Film B, only the rocket plants not treated with UV-B showed values significantly different from all other UV-B treatments of the same film. Mormile et al. [5] demonstrated that by using a greenhouse plastic film with 27% UV-B transmittance for rocket salad cultivation, it is possible to increase the nutraceutical elements (chlorophylls and carotenoids) in comparison with the same species grown under film blocking such radiation. UV-B exposure was also observed to increase the leaf carotenoid content in tobacco plants [31] and in bell pepper [45].

### 3.3. Total Phenolic Content and Phenolic Profile of Rocket Leaves

The interaction between the greenhouse cover films and UV-B application significantly affected the phenolic content (Figure 5), while the phenolic profile was affected only by the single effect of greenhouse cover films (Table 1).

The total phenolic content ranged from 321.16 to 586.03 mg GAE/100 g D.W. (Figure 5) and was generally higher in samples grown under Film C than in the others. These values were lower compared to those found by Heimler et al. [46], which showed a polyphenol content of 100 mg GAE/100 g fresh weight, and to those found by Koukounaras et al. [47], which showed a concentration ranging from 79 mg GAE/100 g fresh weight to 110 mg GAE/100 g fresh weight in *Eruca sativa* treated with different temperature times. According to Schreiner et al. [14], UV-B radiation treatments generate secondary metabolites, such as phenolic compounds and carotenoids, as a plant defence response against harmful UV-B radiation.

Again, under Film C, higher values of phenolics were reached, but the increases were more moderate: 33.3% and 8.2% over Film A and Film B, respectively (Figure 5). Moreover, the best performance for this parameter was recorded in leaves of rocket treated with UV-B for 45 s. Notably, under the other two films, the differences between the UV-B treatments were minimal.

The predominant individual phenolic compounds detected were kaempferol, isorhamnetin, and quercetin (in the range 36.41–3155.51, 250.35–953.80, and 146.02–422.80 mg/100 g D.W., respectively), while rutin, caffeic acid, and gallic acid were found at concentrations below 80 mg/100 g D.W. (Table 1). These compounds were also found with similar values by Pasini et al. [9] in 32 cultivars of *Eruca sativa* and 5 of *Diplotaxis tenuifolia*. Kaempferol, the most abundant phenolic compound we have found, has pharmacological activity such as antioxidant, anti-inflammatory, antimicrobial, anticancer, cardioprotective, antidiabetic, analgesic, and antiallergic activities [48]. Quercetin and isorhamentin also have biological activities such as antioxidant and anti-inflammatory (quercetin), antiplatelet, and anticoagulant activity (isorhamentin) [49,50]. Jaramillo et al. [51] reported that the bioavailability of isorhamnetin (3′-*O*-methylquercetin) is higher than that of quercetin, because the absorption and metabolic stability of methylated flavonoids are dramatically increased when compared with unmethylated parent molecules. The content of quercetin was higher than the content of quercetin in Durazzo et al. [12], which showed a concentration in the range 0.50–7.74 mg/100 g of fresh weight depending on the cultivation practices used. Jin et al. [52] reported that the content of quercetin, kaempferol, and isorhamentin ranged from 100–1200 mg/100 g D.W., 1400–2500 mg/100 g D.W., and 10–350 mg/100 g D.W., respectively, in rocket grown with different light levels during the cultivation period.

For all phenols individuated with HPLC analysis, the best performance was recorded in plants grown under Film C, which was always significantly different from the values of plants grown under Film A, except for gallic acid, for which the lowest values were recorded under Film B (Table 1). Regarding caffeic acid, the Film C value was approximately eight times more than the mean value of Film A and Film B. Finally, no differences between Film C and Film B were recorded for kaempferol, isorhamnetin, quercetin, and rutin, which on average elicited 87.2%, 52.8%, 46.5%, and 94.7% increases over Film A, respectively (Table 1).

### 3.4. Antioxidant Activity and Ascorbic Acid Content of Rocket Leaves

The antioxidant activity was measured by the two most common radical scavenging assays using ABTS and DPPH (Table 2). The values ranged from 1276 to 2573 and from 988 to 1821 mg TE/100 g D.W. in the ABTS and DPPH assays, respectively, and they were in agreement with those found by Araújo-Rodrigues et al. [41]. ABTS values were always higher than DPPH, as also found by Floegel et al. [53], who concluded that the ABTS assay better estimates the antioxidant capacity of foods, particularly vegetables. According to this study, the ABTS assay is strongly correlated with the phenolic and flavonoid content and should be preferred when applied to a variety of plant foods containing hydrophilic, lipophilic, and highly pigmented antioxidant compounds.

The ascorbic acid content (Figure 6) was in the range 374–1199 mg/100 g D.W. and was the most abundant antioxidant compound found in rocket samples. These findings were consistent with the literature data [54]. Szwejda-Grzybowska et al. [55] showed a concentration ranging from 681 to 1148 mg/100 g D.W. depending on the time of storage and previous treatment, while Ragusa et al. [39] showed a concentration that ranged between 470 mg/100 g D.W. and 610 mg/100 g D.W. depending on the germination temperature. The dynamic relationship between ascorbic acid and reactive oxygen species (ROS) and their participation in numerous metabolic and cell signalling processes have been well demonstrated. In fact, as a major component of the ascorbate-glutathione cycle, ascorbic acid helps to modulate oxidative stress in plants by controlling ROS detoxification [54].

The interaction between greenhouse cover films and UV-B application affected both the antioxidant activity measured by DPPH and ascorbic acid content (Table 2, and Figure 6); instead, the antioxidant activity measured by ABTS was affected only by greenhouse cover films (Table 2). The lipophilic antioxidant activity was elicited by Film C, which showed a more than 80% increase over Film A and an 18.2% increase compared to Film B, which, in turn, was approximately 53% higher than Film A (Table 1).

Additionally, for the DPPH antioxidant activity, higher values were recorded under Film C; in particular, the treatment with UV-B for 660 s was higher than all other treatments and significantly different from them. Under this film, no differences, instead, were recorded between the other UV-B treatments. The lowest values were recorded under Film A, with the plants not treated with UV-B not different from those treated for 45 s (Table 2).

Regarding the ascorbic acid content, the trend was similar to the trend of the two previous parameters, with Film C eliciting 77.8% and 45.5% increases over Film A and Film B, respectively (Figure 6). In addition, once again, the treatment with UV-B for 660 s under Film C was higher, and the Film A-I treatment was lower than all other treatments and significantly different from them. Notably, under Film C, all UV-B treatments were different, with an increasing trend at the increased duration of UV-B application.

## 4. Conclusions

The findings of the current research highlighted an improvement in the nutritional quality of the wild rocket (*Diplotaxis tenuifolia* L.). When rocket is cultivated under a tunnel covered by a diffuse light film with a window for UV-B, it is treated with this type of radiation through an artificial source in the postharvest phase. Specifically, the bioactive compounds (total phenols, carotenoids, and ascorbic acid), well known for their beneficial effects on human health and the antioxidant activity measured by DPPH, as well as chlorophyll a and b, have been increased by the combination of this diffuse light film able to transmit a well-calibrated quantity of UV-B radiation and a postharvest UV-B treatment. In particular, for most of the analysed parameters, the treatment with an energy dose of 0.2 KJ m^−2^ (45 s) was sufficient to improve the quality. Moreover, the greenhouse light-diffused film with a window for UV-B also increased the dry matter percentage, antioxidant activity measured by the ABTS method and kaempferol, isorhamnetin, quercetin, rutin, caffeic acid, and gallic acid.

Therefore, the nutritional quality of the rocket (*Diplotaxis tenuifolia* L.), subjected to specific treatments with well-calibrated UV-B radiation, would seem to be improved, with a greater effect of the greenhouse cover film.

Thus, this work provides important findings for the choice of greenhouse plastic film (diffused light film, clear plastic film, or diffused light film with a tailored UV-B transmission dose) and for the choice of UV-B radiation postharvest treatment to apply.

However, further research is needed because it is possible that the response to UV-B radiation is species-specific, especially regarding the individuating of the right dose to use for growing wild rocket salad.

## Figures and Tables

**Figure 1 foods-11-04093-f001:**
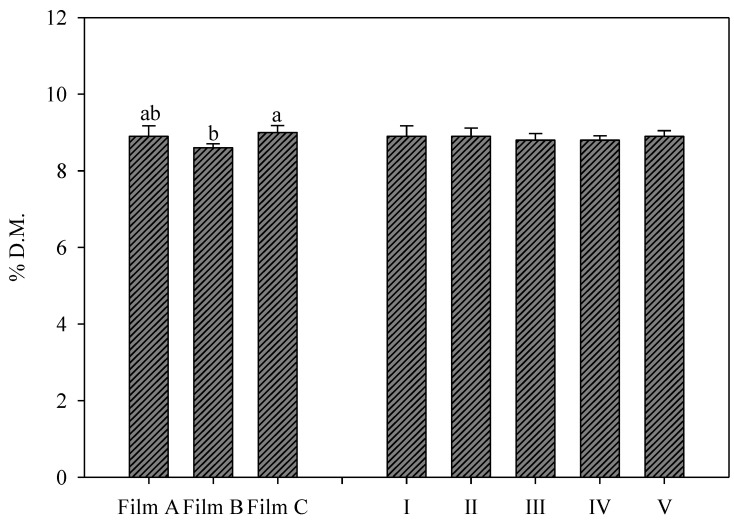
Dry matter percentage (% *w*/*w*) in wild rocket leaves as affected by greenhouse cover film (Film A: diffused light film; Film B: clear film; Film C: diffused light film with UV-B window). Vertical bars indicate standard error; different letters indicate significant differences according to Tukey’s test (*p* ≤ 0.05); ns indicates no significant differences.

**Figure 2 foods-11-04093-f002:**
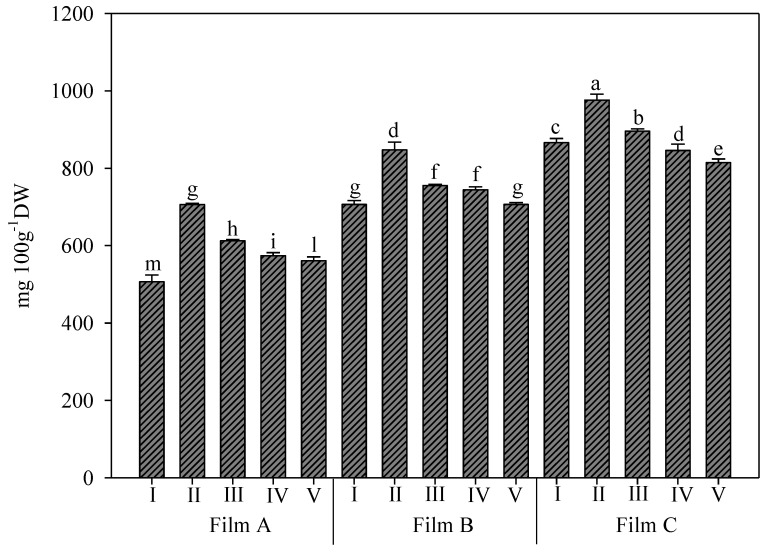
Chlorophyll a content (mg/100 g of dry weight D.W.) in wild rocket leaves as affected by interaction between greenhouse cover film (Film A: diffused light film; Film B: clear film; Film C: diffused light film with UV-B window) and UV-B applications (I: not treated with UV-B; II: treated with UV-B for 45 s; III: treated with UV-B for 150 s; IV: treated with UV-B for 330 s; V: treated with UV-B for 660 s). Vertical bars indicate standard error; different letters indicate significant differences according to Tukey’s test (*p* ≤ 0.05).

**Figure 3 foods-11-04093-f003:**
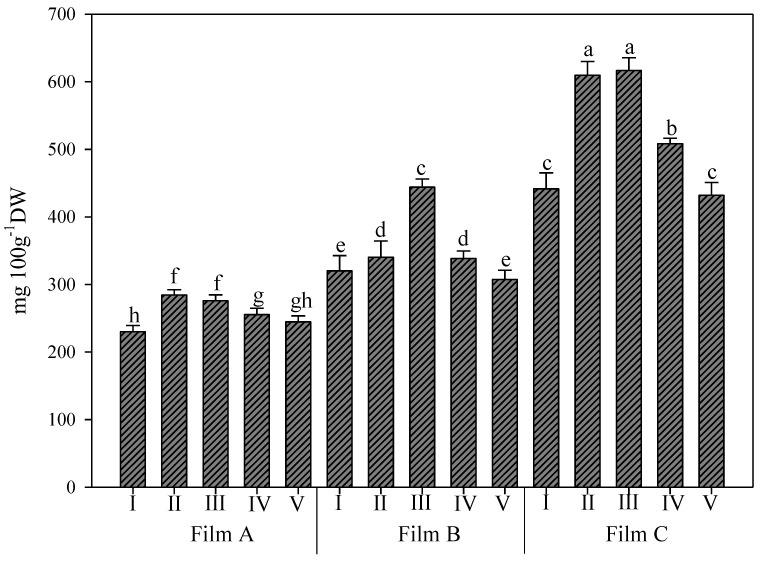
Chlorophyll b content (mg/100 g of dry weight D.W.) in wild rocket leaves as affected by interaction between greenhouse cover film (Film A: diffused light film; Film B: clear film; Film C: diffused light film with UV-B window) and UV-B applications (I: not treated with UV-B; II: treated with UV-B for 45 s; III: treated with UV-B for 150 s; IV: treated with UV-B for 330 s; V: treated with UV-B for 660 s). Vertical bars indicate standard error; different letters indicate significant differences according to Tukey’s test (*p* ≤ 0.05).

**Figure 4 foods-11-04093-f004:**
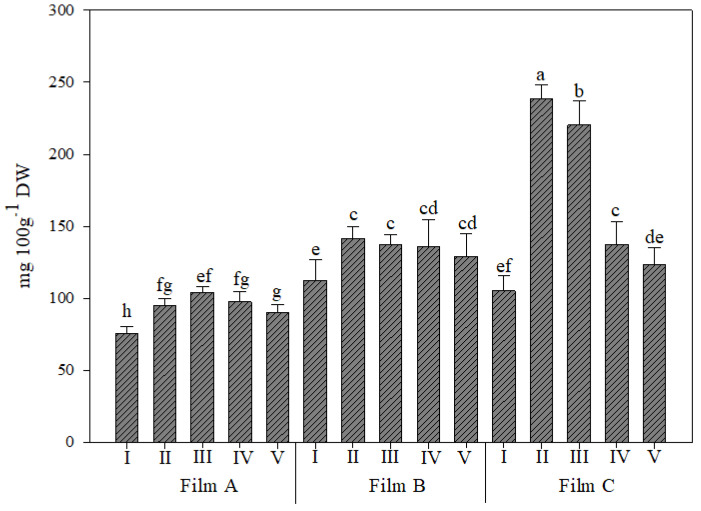
Carotenoid content (mg/100 g of dry weight D.W.) in wild rocket leaves as affected by interaction between greenhouse cover film (Film A: diffused light film; Film B: clear film; Film C: diffused light film with UV-B window) and UV-B applications (I: not treated with UV-B; II: treated with UV-B for 45 s; III: treated with UV-B for 150 s; IV: treated with UV-B for 330 s; V: treated with UV-B for 660 s). Vertical bars indicate standard error; different letters indicate significant differences according to Tukey’s test (*p* ≤ 0.05).

**Figure 5 foods-11-04093-f005:**
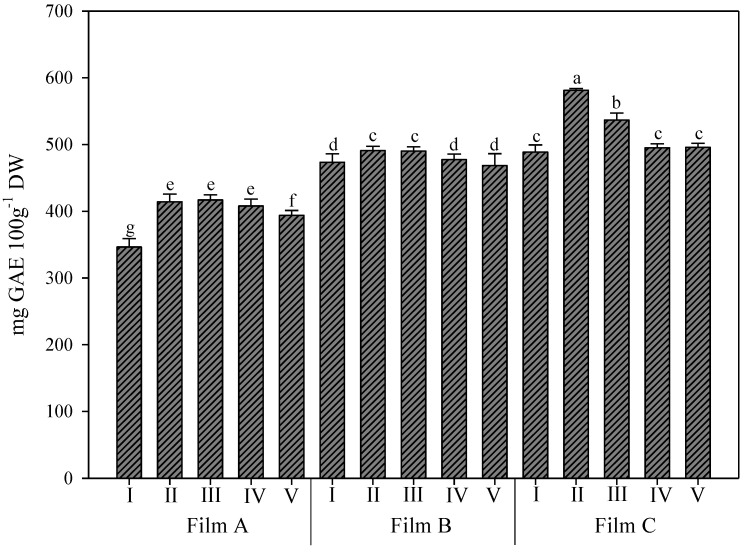
Total phenolic content (mg gallic acid equivalent GAE/100 g of dry weight D.W.) in wild rocket leaves as affected by interaction between greenhouse cover film (Film A: diffused light film; Film B: clear film; Film C: diffused light film with UV-B window) and UV-B applications (I: not treated with UV-B; II: treated with UV-B for 45 s; III: treated with UV-B for 150 s; IV: treated with UV-B for 330 s; V: treated with UV-B for 660 s). Vertical bars indicate standard error; different letters indicate significant differences according to Tukey’s test (*p* ≤ 0.05).

**Figure 6 foods-11-04093-f006:**
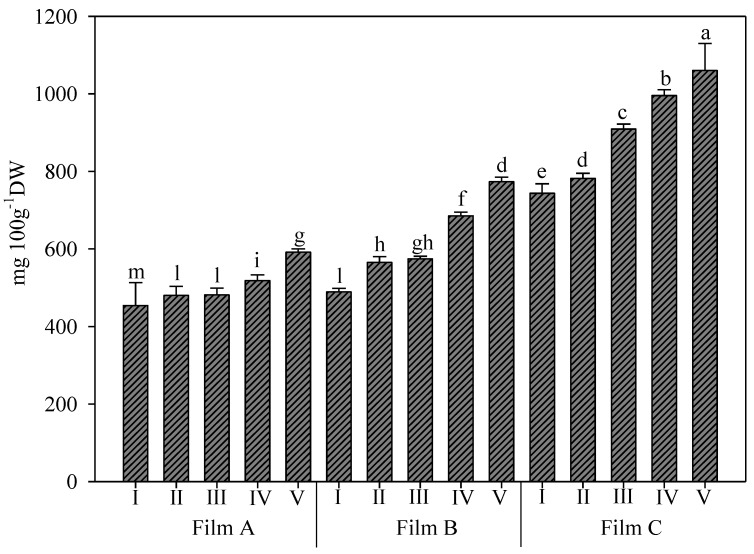
Ascorbic acid content (mg/100 g of dry weight D.W.) in wild rocket leaves as affected by the interaction between greenhouse cover film (Film A: diffused light film; Film B: clear film; Film C: diffused light film with UV-B window) and UV-B application (I: not treated with UV-B; II: treated with UV-B for 45 s; III: treated with UV-B for 150 s; IV: treated with UV-B for 330 s; V: treated with UV-B for 660 s). Vertical bars indicate standard error; different letters indicate significant differences according to Tukey’s test (*p* ≤ 0.05).

**Table 1 foods-11-04093-t001:** Phenolic compound content (mg/100 g D.W.) in wild rocket leaves as affected by the main effect of greenhouse cover film (Film A: diffuse light film; Film B: clear film; Film C: diffused light film with UV-B window), and UV-B applications (I: not treated with UV-B; II: treated with UV-B for 45 s; III: treated with UV-B for 150 s; IV: treated with UV-B for 330 s; V: treated with UV-B for 660 s).

Treatments	Kaempferol	Isorhamnetin	Quercetin	Rutin	Caffeic Acid	Gallic Acid
Greenhouse film						
Film A	771.3 ^b^	386.1 ^b^	220.8 ^b^	26.4 ^b^	5.2 ^b^	26.0 ^a^
Film B	1244.4 ^a^	566.6 ^a^	341.5 ^a^	49.3 ^a^	6.0 ^b^	14.4 ^b^
Film C	1643.5 ^a^	613.5 ^a^	305.5 ^a^	53.5 ^a^	43.6 ^a^	23.5 ^a^
UV-B						
I	1366.3	529.0	296.0	48.2	10.6	22.4
II	1273.4	536.8	290.5	38.5	25.6	27.3
III	971.6	456.8	266.7	33.6	11.2	21.0
IV	1368.4	543.1	301.3	45.6	13.3	16.3
V	1119.1	544.5	291.8	49.3	30.6	19.3
Significance						
Greenhouse film (F)	**	**	**	**	*	**
UV-B (L)	NS	NS	NS	NS	NS	NS
F x L	NS	NS	NS	NS	NS	NS

NS, *, ** nonsignificant or significant at *p* < 0.05 and 0.01, respectively. Different letters within each column indicate significant differences at *p*≤ 0.05.

**Table 2 foods-11-04093-t002:** Antioxidant activity (mg TE/100 g D.W.) in wild rocket leaves as affected by the main effect of greenhouse cover film (Film A: diffused light film; Film B: clear film; Film C: diffused light film with UV-B window) and the interaction between films and UV-B application (I: not treated with UV-B; II: treated with UV-B for 45 s; III: treated with UV-B for 150 s; IV: treated with UV-B for 330 s; V: treated with UV-B for 660 s).

Treatments		ABTS mg TE/100 g D.W.	DPPH mg TE/100 g D.W.
Greenhouse film	UV-B		
Film A	I	1322.8	1074.0
	II	1338.6	1088.3
	III	1348.0	1132.1
	IV	1287.1	1023.2
	V	1375.4	1148.8
Mean		1334.4 ^c^ ± 32.6	1093.3 ^c^ ± 49.7
Film B	I	1965.9	1235.2
	II	1975.3	1210.1
	III	2038.0	1301.6
	IV	2074.2	1264.0
	V	2148.7	1286.5
Mean		2040.4 ^b^ ± 75.3	1259.4 ^b^ ± 37.3
Film C	I	2303.8	1405.3
	II	2373.5	1586.2
	III	2394.8	1613.7
	IV	2455.6	1620.9
	V	2529.9	1756.0
Mean		2411.5 ^a^ ± 85.6	1596.4 ^a^ ± 125.5
Significance			
Greenhouse Film (F)		**	**
UV-B (L)		NS	**
F x L		NS	**

NS, ** Not significant or significant at *p* < 0.05 and 0.01, respectively. Different letters within each column indicate significant differences at *p*≤ 0.05.

## Data Availability

Data is contained within the article and Appendix A.

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
