# Peer review of "Bioactive Compounds and Antioxidant Properties of Wild Rocket (Diplotaxis Tenuifolia L.) Grown under Different Plastic Films and with Different UV-B Radiation Postharvest Treatments"

_foods, 2022, doi:10.3390/foods11244093_

Round 1

Reviewer 1 Report

The manuscript entitled “Bioactive compounds and antioxidant properties of wild rocket (Diplotaxis tenuifolia L.) grown under different plastic films and with different UV-B radiation postharvest treatments” by authors Raffaele Romano, Ida Di Mola, Fabiana Pizzolongo, Lucia De Luca, Eugenio Cozzolino, Massimo Rippa, Lucia Ottaiano, Pasquale Mormile and Mauro Mori, represents an interesting paper describing the influence of UV-B radiation on the content of secondary metabolites, mostly antioxidants (flavonoids), as a plant protectors, i.e. plant response, against ROS generated by UV-B radiation. This work provides important findings for the choice of greenhouse plastic film and for the choice of UV-B radiation postharvest treatment to apply in order to obtain higher quantities of antioxidants. 

General remarks

Make uniform units “L” of “l” for litar throught the text

Write “s” instead of “second” and “sec” throught the text

When authors cite the results of other authors, the first author’s last name of the cited reference should be written before the reference number, for example line 189: “by Romano et al. [38]” instead of “by [38]”.

The innovation of the article was enough. Measurement wavelenghts have been given in the reference cited, but they can be mentioned again. The hplc chromatograms may be provided.

Results and discussion

Line 337: Write “Eruca sativa” and “Diplotaxis tenuifolia” italic

Line 340: Few sentences about bioactivity of quercetin and isorhamnetin should be added in discussion, since these two flavonoids have been found in significant amounts along with kaempferol.

Line 341: Write “the” instead of “thr”

References are relevant, but not mostly within last 5 years.

Coclusion

Line 426: Write “Diplotaxis tenuifolia” italic

Author Response

We would like to thank the reviewer for careful and thorough reading of this manuscript, which helped to improve the quality of this manuscript. Our responses follow in red in the attached file.

Reviewer 2 Report

 In this manuscript,  the production of nutritional elements in wild rocket (Diplotaxis tenuifolia L.) leaves treated with  exposure to UV-B radiation was measured.

1. More data descriptions should be added in the abstract ;

2. Novelty of this article need to be carefully refined and written in the introduction. At present, it seems that the innovation of the article is not enough;

3. In the experiment design, the soil of greenhouses with film A and film B is the same, but the soil of greenhouse with film C is different from the former two, so it is difficult to explain whether the difference in  results is caused by the different film or the difference in soil composition;

4. There are many irregularities in the way of citing references in the full text, which need to be revised;

5. All of  % must be clearly stated whether it is W/W or V/V in the full text; 

6. Is the unit of the homogenizer *g? Should it be rpm?

7. What are the measurement wavelengths for chlorophyll a, b and carotene in the method?

8. The HPLC chromatograms of standards and samples need to be provided for the determination of phenolic acids;

9. The data in the data table should be expressed as Mean±S.D.;

10. There are  some obvious grammatical errors in the text, which need to be corrected by the native speaker.

Author Response

We would like to thank the reviewer for the thoughtful comments and constructive suggestions, which helped to improve the quality of this manuscript. Our responses follow in red in the attached file.

Round 2

Reviewer 2 Report

In the revised version, the innovation of the manuscript and the standardization of writing have made great progress.